# Taphonomic and Diagenetic Pathways to Protein Preservation, Part I: The Case of *Tyrannosaurus rex* Specimen MOR 1125

**DOI:** 10.3390/biology10111193

**Published:** 2021-11-17

**Authors:** Paul V. Ullmann, Kyle Macauley, Richard D. Ash, Ben Shoup, John B. Scannella

**Affiliations:** 1Department of Geology, Rowan University, Glassboro, NJ 08028, USA; macauleyk4@students.rowan.edu; 2Department of Geology, University of Maryland, College Park, MD 20742, USA; rdash@umd.edu; 3Absaroka Energy & Environmental Solutions, Buffalo, WY 82834, USA; ben.shoup@absarokasolutions.com; 4Museum of the Rockies, Montana State University, Bozeman, MT 59717, USA; john.scannella@montana.edu; 5Department of Earth Sciences, Montana State University, Bozeman, MT 59717, USA

**Keywords:** REE, *Tyrannosaurus rex*, molecular paleontology, geochemical taphonomy, diagenesis, bone, protein, collagen, Hell Creek Formation

## Abstract

**Simple Summary:**

Contrary to traditional views, fossil bones have been shown to occasionally retain original cells, blood vessels, and structural tissues that are still comprised, in part, by their original proteins. To help clarify how such remarkable preservation occurs, we explored the fossilization history of a famous *Tyrannosaurus rex* specimen previously shown to yield original cells, vessels, and collagen protein sequences. By analyzing the trace element composition of the femur of this tyrannosaur, we show that after death its carcass decayed underwater in a brackish, oxic, estuarine channel and then became buried by sands that quickly cemented around the bones, largely protecting them from further chemical alteration. Other bones yielding original proteins have also been found to have fossilized within rapidly-cementing sediments in oxidizing environments, which strongly suggests that such settings are conducive to molecular preservation.

**Abstract:**

Many recent reports have demonstrated remarkable preservation of proteins in fossil bones dating back to the Permian. However, preservation mechanisms that foster the long-term stability of biomolecules and the taphonomic circumstances facilitating them remain largely unexplored. To address this, we examined the taphonomic and geochemical history of *Tyrannosaurus rex* specimen Museum of the Rockies (MOR) 1125, whose right femur and tibiae were previously shown to retain still-soft tissues and endogenous proteins. By combining taphonomic insights with trace element compositional data, we reconstruct the postmortem history of this famous specimen. Our data show that following prolonged, subaqueous decay in an estuarine channel, MOR 1125 was buried in a coarse sandstone wherein its bones fossilized while interacting with oxic and potentially brackish early-diagenetic groundwaters. Once its bones became stable fossils, they experienced minimal further chemical alteration. Comparisons with other recent studies reveal that oxidizing early-diagenetic microenvironments and diagenetic circumstances which restrict exposure to percolating pore fluids elevate biomolecular preservation potential by promoting molecular condensation reactions and hindering chemical alteration, respectively. Avoiding protracted interactions with late-diagenetic pore fluids is also likely crucial. Similar studies must be conducted on fossil bones preserved under diverse paleoenvironmental and diagenetic contexts to fully elucidate molecular preservation pathways.

## 1. Introduction

### 1.1. Biomolecular Preservation in Fossils

Preservation of endogenous biomolecules like proteins and DNA in ancient fossils was once thought implausible by many paleontologists. The plethora of microbial and inorganic agents of decay inherent in fossilization were long expected to inevitably either breakdown labile biomolecules into minute and useless components, or alter them so thoroughly via recombination and crosslinking that they become intractably unrecognizable [1]. Yet, as discussed in detail by several recent reviews [2,3,4,5,6,7], innovative adaptations of molecular biology techniques toward analyzing fossil samples and significant advances in analytical resolution have proven that recognizable biomolecules persist in many, diverse fossils. Ancient DNA and even entire nuclear and mitochondrial genomes are now being recovered from Pleistocene fossils [8,9,10,11,12], some of which are older than 1 My [13], and proteins such as collagen I, hemoglobin, and β-keratin have been identified within vertebrate fossils dating back to the Jurassic [14,15,16,17,18,19,20,21]. Collagen I peptides have even been recovered from two Cretaceous non-avian dinosaur bones [18,22,23], and cladistic analyses found those peptides to place non-avian dinosaurs in their expected phylogenetic context within Archosauria [18,24], thus corroborating their authenticity. The growing number of these reports by independent research groups using diverse analytical techniques clearly demonstrates that ‘surviving’ fossilization is not an insurmountable challenge for biomolecules. Yet, this concept remains controversial, in part because there are still many gaps in our understanding of the geochemical processes that result in fossilization in general, let alone the processes that result in exceptional preservation of soft tissues and their component molecules.

“Traditional” explanations attributing the preservation of proteinaceous tissues such as skin, blood vessels, or feathers to simple carbonization (e.g., [25]) or mineral replacement (e.g., via phosphatization [26]) are clearly insufficient, as they cannot explain cases of retention of endogenous molecular signatures in such fossil tissues (e.g., [15,27]). Yet, it is only within the last decade that molecular paleontologists have begun developing alternative hypotheses about preservation mechanisms capable of fostering long-term molecular stability. Schweitzer et al. [28,29] were the first to propose such a mechanism, suggesting that iron-catalyzed free radical reactions could mediate natural tissue fixation by inducing intramolecular crosslinking. Ensuing actualistic experiments by Boatman et al. [30] found support for this hypothesis in that Fenton- and glycation-treated modern chicken collagen samples exhibited the same types of crosslinks present in the walls of fossil blood vessels recovered from a Cretaceous *Tyrannosaurus rex* femur (MOR 555/USNM 555000). To date, only two other studies have directly addressed preservation mechanisms. Schweitzer et al. [31] suggested that concurrent recrystallization of bone hydroxyapatite may encase the molecular condensation products described above as they form, shielding them in a manner similar to intracrystalline proteins (cf. [32]), whereas Edwards et al. [33] alternatively suggested that ternary complexation of biomolecules with dissolved metal cations and mineral crystal surfaces may stabilize them over geologic timescales. As the brevity of this summary shows, attempts to investigate mechanisms of molecular preservation remain rare.

To better elucidate these (and other, as-yet-unrecognized) potential preservation mechanisms, it is imperative to identify geochemical regimes conducive to such “exceptional” preservation and characterize the full suite of physicochemical taphonomic parameters at play. Although certain taphonomic circumstances can reliably be linked to “exceptional” preservation (e.g., rapid burial) [34,35], many geologic and geochemical variables need further study to elucidate their influence(s) on decay at the molecular level. For example, how much do factors such as groundwater chemistry and diffusion history control the preservation potential of biomolecules? Are the depositional setting (e.g., floodplain, seafloor) and geochemical microenvironment of burial primary factors controlling whether or not original soft tissues and their component molecules persist? If so, which sedimentary facies and diagenetic histories are most conducive to biomolecular preservation in fossils, and why? These important questions remain largely unexplored.

### 1.2. Trace Element Taphonomy of Fossil Bone

Among the many established and emerging analytical methods that could be used to shed light on potential answers to the questions listed above, trace element analyses are arguably the most useful, as they offer unparalleled windows into the geochemical and diagenetic history of fossils. Rare earth elements (REE; lanthanum-lutetium), uranium, scandium, and other trace elements released from sediments into groundwaters are ubiquitously adsorbed by bone hydroxyapatite during fossilization (primarily via surface adsorption and cationic exchange into the crystal lattice, leaving histologic structure unaltered; [36,37]). These elements are negligibly present in bone mineral during life, meaning that their presence in a fossil bone reflects the geochemical and hydrodynamic history of the diagenetic environment [37,38]. The proportions of trace elements adsorbed depends largely on groundwater chemistry [37], and their spatial distributions within fossil bone tissues form records of the pore-fluid interactions a bone experienced through diagenesis [36,39,40,41,42,43,44]. In short, analyses of REE and other trace elements offer unique insights not only into the magnitude of chemical alteration a specimen has undergone, but also: (1) the chemistry of ancient surface and groundwaters in the burial environment (e.g., [43,45,46]); (2) spatial and temporal trends in redox conditions both within and around biologic remains as they were fossilizing (e.g., [46,47,48]), and; (3) the number and timing of interactions a specimen had with pore fluids through diagenesis [39].

Given these diverse and critically-relevant utilities, two of us (P.V.U. and R.D.A.) elected to employ trace element analyses in parallel with molecular assays in an initial case study examining the diagenetic history of *Edmontosaurus* bones retaining endogenous collagen I from a mass-death bonebed in the Cretaceous Hell Creek Formation [20,46,49,50]. Although those studies clarified one set of paleoenvironmental, diagenetic, and geochemical circumstances conducive to biomolecular preservation, all of the specimens examined in that project derived from a single locality and taxon. The current study builds upon our prior work by examining the diagenetic history of another non-avian dinosaur preserved under drasticallydifferent paleoenvironmental and diagenetic circumstances within the Hell Creek Formation: *Tyrannosaurus rex* specimen MOR 1125. This specimen became one of the most widely-known fossils in the world when Schweitzer et al. [51,52] and Asara et al. [22] reported, respectively, the preservation of original bone cells, blood vessels, and pliable proteinaceous matrix in its right femur and both tibiae, as well as endogenous collagen I peptides in its right femur. Those studies of MOR 1125 ignited unprecedented interest in the now-growing field of molecular paleontology, so it is due time for the taphonomic and diagenetic history of this specimen to be resolved in comprehensive detail.

## 2. Taphonomic and Geologic Context

MOR 1125 was collected from exposures of the Maastrichtian Hell Creek Formation northwest of Jordan, MT, and just south of the Fort Peck Reservoir on lands managed by the United States Fish and Wildlife Service (Charles M. Russell Wildlife Refuge) (Figure 1). Its more than 220 skeletal elements were disarticulated but closely associated (Figure 2) within sandstone underneath ~50 ft of overburden (Figure 3A and Appendix A). All preserved cranial elements were found within a 5 m^2^ area in the northern corner of the quarry, and nearly all teeth were found dislodged from the jaw elements in two clusters adjacent to the skull bones. Based on data from the 25 m^2^ excavated in 2002 (Figure 2), skeletal element abundances range from 0 to 16 specimens/m^2^ with an average of 5 specimens/m^2^. Field data collected by Museum of the Rockies crews reveal that the bones were stratigraphically separated from one another by as much as 55 cm, though all of the cranial elements were found within a narrower ~15 cm interval. Although this stratigraphic interval is thick enough to accommodate bones stacked on top of one another, almost all specimens were found spatially isolated. Strike measurements acquired for 14 long bones (e.g., limb bones, ribs) during the 2002 field season show a bimodal pattern comprising a northeast-southwest trend and a nearly north-south trend (Figure 3B), implying probable hydraulic orientation of bones within the quarry. Bones range in size from phalanges and cervical ribs up to girdle elements and large limb bones (i.e., bones pertaining to both Voorhies Groups I and II were recovered; cf. [53]), and all major portions of the body are represented. Most of the bones are complete, though some exhibit transverse and longitudinal fractures stemming from minor post-fossilization compaction. None of the bones exhibit any noteworthy signs of weathering or abrasion, and the femur examined by Schweitzer et al. [51,52] and Asara et al. [22] exhibits infilling of the medullary cavity by crushed trabeculae and sedimentary matrix.

Schweitzer et al. [51] (p. 1953) briefly characterized the lithology from which MOR 1125 was recovered as a “soft, well-sorted sandstone that was interpreted as estuarine in origin”, but the stratigraphy and sedimentology of the quarry have never been reported in detail, despite their relevance to interpreting the depositional environment and taphonomic history of this specimen. Below, we briefly summarize the stratigraphy of the quarry and the entire butte encasing this specimen.

The specimen was recovered from the Basal Sand of the Hell Creek Formation (*sensu* Hartman et al. [56] and Fowler [54]), 1.5 m above its basal contact with underlying Colgate tidal flat facies. This places it within the lower unit (L3 *sensu* Horner et al. [57]) of the Hell Creek Formation (Appendix A). Dark, marine deposits of the Bearpaw Shale are exposed near the base of the butte from which the specimen was recovered, and they exhibit a gradational contact with a shallow-marine sequence identified as the Fox Hills Sandstone. The Fox Hills Formation is approximately 11.2 m thick and coarsens subtly up-section from silty, hummocky cross-stratified fine sands at the base to low-angle planar cross-stratified, fine-medium sands near the top (Appendix A). Sublitharenitic, trough cross-stratified, fine to medium-grained sands of the Colgate Sandstone erosively scour into the Fox Hills Formation (Appendix A), as is common across the region (following the stratigraphic definitions of Fowler [54]).

Specifically, MOR 1125 was found entombed near the base of a normally-graded, swaley to trough cross-bedded channel lag deposit within a 11.8 m thick section of fine to coarse-grained, trough cross-bedded sandstones (Figure 3A and Appendix A). Pebbles and rounded, silty, rip-up clasts occur immediately beneath the bones, and all three of these larger clast types are supported by a homogenous matrix of medium and coarse sand, all of which are indicative of deposition within a relatively high-energy channel (see Discussion).

Strata exposed in the upper portion of the quarry wall largely consist of cross-bedded fluvial channel sandstones, shaly floodplain/overbank mudstones, and massive crevasse-splay sandstones typical of the middle portion of the Hell Creek Formation [54,55,58,59,60,61]. For further details on the sedimentology of the entombing sandstone and strata overlying it, please see Appendix A.

## 3. Materials and Methods

### 3.1. Materials

A portion of cortex excised from the midshaft of the right femur of *Tyrannosaurus rex* MOR 1125 was used in this study. Although it is not the exact piece of cortex from which Schweitzer et al. [52] and Asara et al. [22] recovered endogenous protein and peptide sequences, respectively, this fragment is derived from the same region of the midshaft of the same bone, thereby minimizing any potential chemical heterogeneities between the two cortex samples. The fragment comprises nearly the entire cortical thickness of the bone, including the intact external cortex margin but not the internal wall of the medullary cavity (hence, none of the medullary tissue identified by Schweitzer et al. [62] is present in this fragment). Macroscopically, the femur is well-preserved, exhibiting no signs of pre-burial weathering or postmortem abrasion.

### 3.2. Methods

#### 3.2.1. Sample Preparation

An autoclaved chisel was initially used to isolate a smaller piece of the midshaft fragment of the femur of MOR 1125 for embedding and sectioning. The resulting subsample, which captured the cortical width of the femur, was then embedded under vacuum in Silmar 41^TM^ resin (US Composites). A Hillquist SF-8 trim saw was used to cut a thick section (~3 mm) from the embedded subsample, which was then rinsed with distilled water and allowed to thoroughly dry. For laser ablation-inductively coupled plasma mass spectrometry (LA-ICPMS) analyses, the thick section was placed directly in the laser ablation chamber; no further polishing was necessary or performed.

#### 3.2.2. LA-ICPMS Analyses

We employed the same mass spectrometry methods as Ullmann et al. [46] in this study, and refer the reader to that publication for details. Briefly, LA-ICPMS was used to examine spatial heterogeneity of REE and other pertinent trace elements in the fossil in order to reconstruct the diagenetic history of the specimen and the geochemical regimes to which it was exposed. Iron concentrations are reported in weight percentage (wt. %), while all other concentrations are reported in parts per million (ppm). To enable comparisons to fossil bones from other sites, REE concentrations were normalized against the North American Shale Composite (NASC) using values from Gromet et al. [63] and Haskin et al. [64] (a subscript N denotes shale-normalized values or ratios). Reproducibility, taken as the percent relative standard deviation for all REE in an NIST 610 glass standard, averaged 2% and was at or below 3% for every element except iron (6.6%). For further analytical specifics of the LA-ICPMS runs performed in this study, please see the Appendix A.

## 4. Results

### 4.1. Overall REE Composition

At the specimen level (i.e., considering all transect data combined), the femur of MOR 1125 exhibits a ∑REE of 596 ppm; this value thus represents the average REE content of the cortex (Table 1). Manganese (Mn) and strontium (Sr) concentrations are the highest of all recorded elements (2439 and 2386 ppm, respectively), with concentrations more than double all the other trace elements and more than an order of magnitude higher than all REE (Table 1). The average concentration of yttrium (Y) is also higher than all REE (1102 ppm), and the average scandium (Sc) concentration is very high (83 ppm) compared to fresh bone. At the whole-bone level, light (LREE) and heavy rare earth element (HREE) concentrations are elevated compared to the middle rare earths (MREE, Sm–Gd), indicating fraction among REE occurred during uptake (see below). Though uranium (U) concentrations exhibit an average (38 ppm) higher than other dinosaur bones recently analyzed from the Hell Creek Formation (2–18 ppm) [46], the femur of MOR 1125 exhibits a comparatively lower amount of iron (0.73 wt. %; compared to 1.23–1.76 wt. % in [46]).

### 4.2. Intra-Bone REE Depth Profiles

All REE exhibit steeply declining concentrations with cortical depth, and certain elements exhibit hints of weak secondary diffusion from within the medullary cavity (e.g., Figure 4A; also see Appendix A). Among the REE, cerium (Ce) concentrations are the highest at the cortical margin (~2200 ppm) and thulium (Tm) exhibits the lowest concentrations at the outer cortex edge (~50 ppm). LREE exhibit the steepest declines (on average from ~1300 ppm near the cortical margin to <15 ppm by 1 cm into the cortex; Figure 4A), generally encompassing a decrease of two orders of magnitude, which is clearly indicative of greater uptake in the external cortex than deeper within the bone. MREE concentration profiles are generally intermediate in slope between those of LREE and HREE, and MREE concentrations are so low throughout the middle and internal cortex (<2 ppm) that they frequently encroach on or fall below detection limit (Data S1). HREE exhibit the flattest profiles among the rare earths (e.g., ytterbium [Yb] in Figure 4A), and unlike LREE and MREE, they exhibit increasing concentrations with depth through the internal cortex (again reflective of fractionation during uptake; see below). For example, Yb concentrations rise from ~30–80 ppm in the middle cortex to ~50–150 ppm in the internal cortex (Data S1). Moreover, HREE only decline from an average of ~250 ppm at the cortical margin to ~40 ppm by 1 cm into the cortex, constituting less than an order of magnitude decrease.

Although intermittent spikes in REE concentrations are present in osteonal tissue surrounding Haversian canals (indicative of uptake through vascular systems, e.g., brief Yb spike near 14 mm in Figure 4A), there are no obvious signs of a uniform deflection in elemental profiles reflective of significant double medium diffusion effects (cf. [44]). Instead, most trace element profiles exhibit a distinct plateau and then drop in concentrations at ~4.7 mm into the cortex (Figure 4). A fine, open, diagenetic crack passes diagonally across the laser transect at this depth. As shown in Figure 4A, REE concentrations are uniformly higher in the bone on the external side of this crack than on the internal side of it.

Scandium (Sc) is the only element to exhibit a distinct, broad peak in concentrations within the middle cortex (Figure 4B). Uranium (U) also exhibits a depth profile shape distinct from those of all the other elements examined, characterized by an initial, weak decrease in concentration from the cortical margin followed by a slow, steady increase in concentration throughout the middle cortex to a stable plateau of the highest concentrations within the bone (~40–50 ppm) within the internal cortex (Figure 4B). Unlike REE, Sc, and Y, the profile of U does not exhibit any disruption related to the crack at 4.7 mm. Iron (Fe) exhibits a nearly flat profile (Figure 4C) with comparatively less locallyrestricted spikes in concentrations in the external cortex than strontium (Sr), manganese (Mn), and barium (Ba), each of which exhibit high concentrations throughout the cortex (Figure 4D). Yttrium (Y) exhibits the same general profile shape as the HREE in the bone, including a slight decrease in concentrations near 15 mm and slight increase in concentrations toward the internal end of the transect (Figure 4C), indicative of similar uptake behavior for these elements in the femur of MOR 1125.

### 4.3. NASC-Normalized REE Patterns

Although the external-most cortex of the femur exhibits considerably greater REE enrichment than deeper portions of the cortex (as is common in fossil bones, e.g., [39,65]; Figure 5A), the external 250 μm of the transect still exhibits an NASC-normalized pattern very similar to that of the bone as a whole (compare Figure 5B and Figure 6A), but with lesser relative enrichment in HREE. Both exhibit a modest negative Ce anomaly (visually evident as a downward deflection of the pattern at this element) and relative LREE depletion and HREE enrichment relative to MREE. Relative enrichment in HREE (perhaps resulting from uptake from brackish pore fluids; see Discussion) is also evident in how closely a data point for the whole-bone composition of MOR 1125 plots to the Yb corner of a Nd_N_-Gd_N_-Yb_N_ ternary plot (Figure 5C). In the external 250 µm of the cortex, shale-normalized concentrations range from ~30–100 times NASC values.

A ternary plot of La_N_-Gd_N_-Yb_N_ (Figure 5D) for each individual laser run compiled into the full transect identifies considerable spatial variation in bone composition (i.e., variation exceeds two standard deviations). This is confirmed by a spider diagram of individual laser runs (Figure 6B) which also shows substantial contrasts in the proportions of REE by laser run, with differences primarily corresponding to cortical depth. As can be seen in these figures, the femur of MOR 1125 generally becomes increasingly enriched in HREE relative to LREE and MREE with increasing depth into the internal cortex, with the bone overall shifting from modestly HREE enriched in the external-most cortex to drastically HREE enriched in the internal cortex and inner half of the middle cortex (roughly the inner half of the transect). Proportionally, this trend shifts from roughly one order of magnitude enrichment of HREE relative to LREE in the external cortex to approximately three orders of magnitude in the internal cortex.

Relative to NASC, middle and internal cortex transects generally exhibit roughly equal depletion in LREE and enrichment in HREE (Figure 6B). The internal-most laser run exhibits uniformly higher REE concentrations than the run immediately external to it. Although the laser run with the lowest concentrations, which crosses the outer portion of the internal cortex, exhibits slightly elevated peaks at neodymium (Nd), Gd, and holmium (Ho), there are no distinct signs of tetrad effects (i.e., ‘M’- or ‘W’-shaped shale-normalized patterns; [47] and references therein) in other laser runs (Figure 6B) or in the bone as a whole (Figure 5B; also see Appendix A for further discussion on potential tetrad effects in MOR 1125).

### 4.4. (La/Yb)_N_ vs. (La/Sm)_N_ Ratio Patterns

At the whole-bone level, the fibula of MOR 1125 exhibits an (La/Sm)_N_ value of 0.75 and a (La/Yb)_N_ of 0.04, reflective of substantial HREE enrichment relative to most environmental water samples, dissolved loads, and sedimentary particulates (Figure 7A). In fact, this extremely low (La/Yb)_N_ value places the bone within the compositional range of marine pore fluids and outside the ranges of all the other environmental samples examined (see Appendix A for a breakdown of the literature sources used for environmental samples).

Plotting REE ratios for individual laser runs reveals a consistent pattern of decreasing (La/Yb)_N_ and unchanging (La/Sm)_N_ with increasing cortical depth (Figure 7B). The laser run including the external margin of the bone exhibits an (La/Yb)_N_ value more than two orders of magnitude greater than laser runs across the internal cortex. All laser runs across the internal cortex exhibit (La/Yb)_N_ ratios < 0.003, and those across the middle cortex still remain <0.01. All laser run (La/Sm)_N_ ratios, regardless of cortical depth, range between 0.6–1.0.

### 4.5. REE Anomalies

(Ce/Ce*)_N_, (La/La*)_N_, and La-corrected (Ce/Ce**)_N_ anomalies, which are based on relative proportions of REE, are essentially absent at the outer cortex edge (Appendix A). However, all three of these anomalies fluctuate between positive and negative values across the length of the transect. Gaps in the data for (Ce/Ce*)_N_ and (La/La*)_N_ anomalies become abundant in the internal cortex due to: (1) concentrations of praseodymium (Pr) and Nd falling below lower detection limit in this region of the bone and; (2) occasionally, Nd concentrations are significantly greater than those of Pr (Data S1).

(Ce/Ce*)_N_ anomalies fluctuate positively and negatively from ~0.4–4.0 across the transect, with values often being slightly negative through much of the external and middle cortices but positive in the internal cortex (Appendix A). There is a distinct ~0.2 magnitude rise in (Ce/Ce*)_N_ values just interior to the open crack at a cortical depth of 4.7 mm, where the values appear to remain consistently alike those at the cortical margin for roughly the next 3 mm. When averaged for the entire bone (Table 1), MOR 1125 exhibits a weakly negative (Ce/Ce*)_N_ value of 0.82, which is consistent with the negative inflection of the NASC-normalized pattern at Ce in the whole-bone spider diagram (Figure 5B).

To further aid in differentiating true, redox-related cerium anomalies from apparent anomalies produced by (La/La*)_N_ anomalies, (Ce/Ce*)_N_ values were also plotted against (Pr/Pr*)_N_ values (following [66]). Anomaly values from the inner regions of the cortex occupy a substantially wider range of both (Ce/Ce*)_N_ and (Pr/Pr*)_N_ than those from the external-most 1 mm of the bone (Figure 8), signifying a relatively more heterogeneous composition in the middle and internal cortices. All but one external cortex data point plot near the upper margins of fields 2a and 4b, indicative of slightly positive La and Ce anomalies, whereas internal cortex measurements plot in every field (Figure 8). Relatively few data points plot in fields 2b and 4a, generally indicative of negative La and Ce anomalies; most of these pertain to the internal cortex (and all of those that do not pertain to the middle cortex).

Quantitative calculations of (La/La*)_N_ anomalies and La-corrected (Ce/Ce**)_N_ anomalies confirm these qualitative inferences. (Ce/Ce**)_N_ anomalies are generally slightly positive throughout the external cortex but become slightly negative (<1) in the middle and internal cortex (Appendix A). Although there are numerous data gaps (due to the Pr and Nd concentration factors noted above), this pattern is exemplified by the innermost 7 mm of the internal cortex exhibiting a negative (Ce/Ce**)_N_ average of 0.94. Additionally, fluctuations in (Ce/Ce**)_N_ values across the middle cortex are considerable, encompassing variation of more than three orders of magnitude. At the whole-bone level, the bone exhibits a slightly positive (Ce/Ce**)_N_ anomaly (1.26; Table 1). When plotted by laser run, (Ce/Ce**)_N_ anomalies display a positive correlation with U concentrations (r^2^ = 0.75; Appendix A). (La/La*)_N_ anomalies are commonly positive in the external cortex, and they decrease steadily to negative values in the inner half of the transect (Appendix A). At the whole-bone level, MOR 1125 exhibits a positive (La/La*)_N_ anomaly (average = 2.83; Table 1); however, this value is drastically biased toward readings from the external cortex due to abundant data gaps in the internal cortex (owing to the Pr and Nd concentration factors noted above).

Yttrium/holmium (Y/Ho) ratios are slightly above chondritic (26) [67] near the cortical margin and through most of the external cortex (range ~30–70). Deeper in the bone, Y/Ho anomalies become increasingly positive through the middle cortex, forming a broad peak near 24 mm of ~90–250, then gently decline through the innermost cortex to values of ~60–100 at the end of the transect (Appendix A). When averaged for the entire transect, the Y/Ho anomaly is positive (59; Table 1).

## 5. Discussion

### 5.1. Clarifying MOR 1125′s Paleoenvironmental and Taphonomic Context

Based on the stratigraphic section recorded across the entire butte in the field (Appendix A), MOR 1125 was recovered from strata comprising the lower unit (L3 of [57] or Reid Coulee unit of [56]) of the Hell Creek Formation, specifically from the Basal Sand [57]. This stratigraphic context, which coincidentally makes MOR 1125 one of the oldest (stratigraphically-lowest) *T. rex* specimen known from the Hell Creek Formation [57], implies that the carcass was likely buried in a low-elevation environment relatively close to the coast (i.e., within 5 km) of the Western Interior Cretaceous Seaway (WIKS; cf. [61,68]). This conclusion is empirically supported by the presence of organic-rich inclined heterolithic strata in the entombing sandstone, as well as a large tree trunk and numerous fossil leaves (likely indicative of subaqeous burial within a low-elevation environment). However, the ratio of sediment to organics in this sandstone is too high for it to have been deposited in a persistent swamp (cf. [69,70]), and it also lacks root traces which would be expected in an abandoned channel or marshy environment [71,72]. Indeed, outside of a significant debris flow event, sandstones would not generally be expected to be deposited in quiescent swamps or marshes [71]. Rather, many aspects of the sedimentology of the entombing sandstone indicate that it was deposited under relatively high energy compared to all other strata observed in the quarry, including its considerable thickness (11.8 m), normally-graded structure, medium to coarse-grained matrix, trough and occasional swaley cross-bedding, and inclusions consisting of pebbles and rounded mud rip-up clasts. Moderate stratigraphic dispersal of the bones of MOR 1125 (by up to 55 cm) is also suggestive of burial occurring in a temporally-persistent, relatively high-energy setting.

Collectively, all of these findings strongly suggest that MOR 1125 was buried within an active channel in a lush, low-elevation environment near the coast. Absence of scales from the freshwater gar *Lepisosteus* in the entombing sandstone, combined with the presence of fragmentary turtle remains and an abundance of organic detritus in this stratum, implies deposition in a brackish estuarine setting, as inferred by Schweitzer et al. [51] (p. 1953); we therefore agree with their interpretation of the depositional setting having likely been an “estuarine” channel. This conclusion is also in agreement with those of Flight [73] and Fowler [54], who also interpreted the Basal Sand of the Hell Creek Formation as representing deposition within estuarine channels. The mean dip direction of cosets within these beds (see Taphonomic and Geologic Context above) indicate that the channel primarily flowed toward the southeast, and an average flow in this general direction is also supported by the predominant south-southeast orientation of long bones within the quarry (inferred to be parallel to the current; Figure 3B).

The cause of death of this tyrannosaur remains unknown. However, we view drought to be an unlikely cause due to the absence of red/oxidized paleosols and caliche (cf. [74,75]). Obrution and miring are also unlikely due to the lack of contorted strata, skeletal articulation, and preferential preservation of bones from the tail, hind limbs, and/or pelvis (cf. [76]).

In contrast, the postmortem history of MOR 1125 is much clearer. Taken together, the spatial distribution of the bones and their well-preserved character indicate that decomposition was allowed to take place long enough to result in complete disarticulation of the skeleton, yet also short enough for the bones to avoid weathering. This combination (as well as marked HREE enrichment, see below) implies that a significant phase of decomposition occurred in an oxic, subaqeous environment (cf. [75]), as the weak temperature swings of constant submersion can thwart significant bone weathering [77]. Based on the close association of the skeletal remains, negligible signs of abrasion, and lack of evidence for significant winnowing (i.e., representation of both Voorhies Groups and all major portions of the skeleton, including small and light-weight bones), burial appears to have taken place very close to the site of skeletonization; the remains can thus be considered modestlyparautochthonous. Given the wealth of lush terrestrial habitats across the floodplains recorded by strata of the Hell Creek Formation [61,78], it is likely the tyrannosaur died at a location slightly upstream from the quarry. Finally, occurrence of all bones in a single bed exhibiting slight normal grading of sediments (Figure 3A) signifies that burial was accomplished by a single depositional event characterized by waning flow competency [79].

We thus conclude the following scenario for the burial history of MOR 1125 based on the available geologic and taphonomic data: this *Tyrannosaurus* perished near a fluvial channel close to the coast of the WIKS. The river/stream carried the carcass downstream, during which time the carcass underwent decomposition primarily underwater. Opening of the fluvial channel into a broader estuary diminished flow competency, which caused the carcass to sink to the floor of a likely brackish estuarine channel where its major soft tissues (e.g., skin and muscle) continued to decay, allowing its skeleton to become disarticulated. A significant, brief rise in flow competency, perhaps fueled by a major rain event/flood, entrained abundant coarse sand, siltstone pebbles, and clay rip-up clasts which buried the remains within the normallygraded deposit now seen at the quarry.

### 5.2. Reconstructing the Geochemical History of MOR 1125

Having resolved the biostratinomic history of the carcass, we now discuss insight into its diagenetic history after burial. In particular, our trace element data provide informative clues about the geochemical history of MOR 1125 which allowed it to retain endogenous cells, soft tissues, and collagen I.

The femur of MOR 1125 exhibits surface concentrations of LREE (average ~1300 ppm; Data S1) comparable to those of other dinosaur bones previously analyzed from the Hell Creek Formation (~100–1400) [46], but overall modest ∑REE (596 ppm) compared to most other Cretaceous bones tested to date (Table 2), which have been reported to exhibit ∑REE ranging from 1100 ppm to over 25,000 ppm [36,80,81,82,83]. Average concentrations of Fe (0.73 wt. %), Sr (2386 ppm), and Ba (888 pm) are also low in this specimen compared to other dinosaur bones recently analyzed from the Hell Creek Formation (1.25–1.76 wt. %, ~2300–3700 ppm, and ~1500–2100 ppm, respectively) [46]. Conversely, MOR 1125 possesses (on average) considerably greater concentrations of Y (1102 ppm), Lu (17 ppm), and U (38 ppm) than other bones from the Hell Creek Formation (7–250 ppm, 0.1–3 ppm, and 2–18 ppm, respectively) [46]. Such comparisons are admittedly coarse as these studies span a wide range of taxa, intra-specimen cortical widths, and depositional environments, but they show that (concerning the vast majority of trace elements considered) the femur of MOR 1125 is only modestly chemically altered for its age. This fact alone may largely account for how this fossil has managed to retain original cells, tissues, and peptide sequences [22,51,52]. The relatively low concentrations of REE and other trace elements in the bone may stem, in part, from a combination of limited availability in the regional surface and groundwaters due to complexation with carbonates and humic acids [83,84,85,86,87,88,89,90] and/or partial earlydiagenetic removal from pore fluids by coprecipitation in secondary phosphates within the surrounding sediments [34,91,92,93,94,95] (see Appendix A for further discussion of these potential sequestration processes).

Although REE concentrations in the femur of MOR 1125 steeply decrease with cortical depth, indicative of a single phase of simple diffusive uptake, considerable HREE enrichment is apparent within the middle and internal cortices (Figure 4A). This pattern of elemental signatures, which greatly contrasts trends of relative LREE and MREE enrichment throughout the cortex of other Hell Creek Formation dinosaur bones we have recently analyzed [46], is consistent with protracted uptake from: (1) relatively HREE-enriched lowland waters (such as those in brackish estuaries or tidallyinfluenced river channels) [36,37,96], and/or; (2) diagenetic pore fluids under oxidizing conditions. At the whole-bone level, the femur exhibits an overall positive (Ce/Ce**)_N_ anomaly (1.26) indicative of overall oxidizing diagenetic conditions. However, a (Ce/Ce**)_N_ anomaly is, on average, absent in the middle cortex (Appendix A), and the internal cortex was actually found to exhibit a slightly negative average of 0.94, indicating that REE uptake in the inner regions of the bone primarily occurred under slightly reducing conditions. We interpret this contrast to reflect the development of a locallyreducing microenvironment within the central medullary cavity and other internal regions of the bone during early diagenesis, during which time the bone was externally exposed to oxidizing conditions (cf. [97]). Three additional chemical attributes identify the external early-diagenetic environment as oxidizing: (1) positive (Ce/Ce**)_N_ anomalies in the external cortex (Appendix A; (2) high concentrations of U throughout the cortex (Table 1 and Figure 4B) [98]; and (3) high Sc concentrations in the external cortex [99,100].

Though recent oxidation could also partially contribute to positive (Ce/Ce**)_N_ values in the external cortex, we view it likely that any such contribution is minor for three reasons. First, any such influence would likely disproportionately raise anomaly values only near the outermost edge of the transect, not throughout the entire external cortex (cf. [101]). Second, the bone was collected from beneath 50 feet of rock rather than near the modern ground surface. Finally, the surrounding sediment is wellcemented, which would hinder efficient percolation of modern pore fluids to and around the fossil. For these reasons, we infer that spatial variations in the (Ce/Ce**)_N_ anomaly across the width of the specimen’s cortex reliably record contrasting early-diagenetic redox conditions within and outside of the bone. Part of this conclusion involves primary, early-diagenetic uptake of REE occurring in an oxidizing environment, which is consistent with the coarse grain sizes of the entombing sediment and the estuarine channel setting inferred from the stratigraphy of the quarry (see above). Thus, our geochemical findings independently support our paleoenvironmental interpretations.

The peculiar “offset” in concentration-depth profiles of many elements (e.g., Sc, Y, REE) at ~4.7 mm, characterized by a brief plateau followed by a dramatic drop in concentrations (e.g., Figure 4A,C), clearly appears to result from diagenetic pore fluid flow along a crack cutting across the laser transect at this cortical depth. The plateau of higher concentrations of each of these elements on the external side of the crack is most likely attributable to relatively protracted uptake from externallyderived pore fluids that entered via this conduit and flowed preferentially through the bone tissue exterior to it. Further, (Ce/Ce*)_N_ and (Ce/Ce**)_N_ anomalies each decrease from oxidizing values external to the crack at 4.7 mm to near-zero values interior to it (Appendix A), indicating that this crack created millimeter-scale spatial contrasts in redox conditions within the external cortex during early diagenesis.

Interestingly, the profiles of Fe, Mn, Sr, Ba, and U do not exhibit any disruption related to the crack at 4.7 mm (Figure 4B–D). For Fe, Mn, and Ba, their generally flat, high concentration-depth profiles imply they are primarily incorporated into common, homogeneouslydistributed, secondary mineral phases (e.g., goethite, Mn oxides, barite) [102]. The flat, high concentration-depth profile of Sr presumably reflects its spatiallyhomogenous substitution for Ca ions in bone apatite, as typically seen in bioapatitic fossils [41,103]. The unique profile shape exhibited by U (Figure 4B) indicates that the majority of U was adsorbed from a pore fluid which diffused through the bone either prior to formation of the crack or during late diagenesis (i.e., well after the bone became a stable fossil). The latter of these two hypotheses would imply that substantial late-diagenetic overprinting occurred to the trace element composition of the bone, which seems unlikely given the low average concentrations of most trace elements with diffusivities in bone similar to that of U (~1 × 10^−14^ cm^2^/s [101]; see Table 1 and Data S1), very weak signs of secondary diffusion from within the medullary cavity (Figure 4), and lack of evidence for leaching or secondary REE incorporation phases (Figure 4 and Figure 5A). Atypical placement of MOR 1125 far from the fields occupied by natural fresh waters and those of estuarine and coastal waters in the (La/Yb)_N_ vs. (La/Sm)_N_ plot (Figure 7A) is also inconsistent with incorporation of a significant proportion of the trace element inventory of the bone from a late-diagenetic pore fluid. Thus, most U was likely incorporated into the bone during early diagenesis.

Many chemical attributes of the femur of MOR 1125 suggest that the chemistry of pore fluids percolating through it changed over time through the primary, early-diagenetic phase of diffusive uptake. For example, typical intra-bone fractionation trends (*sensu* [104]) in the (La/Yb)_N_ vs. (La/Sm)_N_ plot for this specimen (Figure 7B, and discussed further in the Appendix A relative to prior literature [40,105,106]) clearly demonstrate that pore fluids became relatively depleted in LREE by the time they reached the middle and internal cortices. Similarly, one would expect elements with similar diffusivities, such as U and REE [101], to develop similarlyshaped concentration depth profiles if pore fluid chemistry was held constant by sustained replenishment, but these elements exhibit blatantlydifferent profile shapes in MOR 1125 (compare Figure 4A,B): whereas REE profiles steeply decline from the cortical margin to low concentrations throughout the interior of the bone, U concentrations gradually increase inward from a subtle minimum near 2–3 mm, such that the highest average concentrations found in the bone occur in the outer portion of the internal cortex (ca. 24 mm). This suggests that the availability of REE for uptake by the bone’s interior dwindled over time while that of U remained relatively high. Such a situation could arise due to the greater mobility of U complexes than those of REE in oxic fluids [83], and the relatively lower partition coefficients (in apatite) of U than REE could further help maintain high U availability in pore fluids diffusing through the internal regions of the bone [101]. The positive correlation between U concentrations and average (Ce/Ce**)_N_ anomaly for each laser run (r^2^ = 0.75; Appendix A) further supports this interpretation, as it suggests that Ce and U were incorporated into the bone over similar timescales [98]. Finally, positive Y/Ho anomalies throughout the middle and internal cortices and positive (La/La*)_N_ anomalies in the external cortex are also likely products of fractionation during uptake from pore fluids which were changing in composition over time (Table 1, and discussed in the Appendix A). If the majority of uptake with fractionation occurred during early diagenesis, as all evidence appears to support, then the composition of MOR 1125 likely reflects uptake from circum-neutral pH groundwaters (cf. [107]).

To summarize, our cumulative trace element data indicate that the femur of MOR 1125 experienced moderate trace element uptake from a circum-neutral pH, HREE-enriched pore fluid during early diagenesis. Decay produced locallyreducing microenvironments within the medullary cavity and in regions within the cortex while it was externally exposed to oxidizing conditions. Combining these findings with the traditional taphonomic observations discussed above (i.e., minimal weathering and abrasion, disarticulation but close association of the remains) reveals that MOR 1125 underwent skeletonization during fairly prolonged, subaqeous decay in a sandy estuary channel and, following burial, its bones remained exposed to an oxic, potentiallybrackish pore fluid through early diagenesis and fossilization, after which they experienced relatively minimal further chemical alteration (Figure 9).

### 5.3. Emerging Taphonomic and Diagenetic Themes

To date, this report constitutes only the second study to have geochemically characterized the depositional circumstances of a pre-Cenozoic locality where bones yield original biomolecules, with the other being recent work by some of us (P.V.U. and R.D.A.) on the Standing Rock Hadrosaur Site (SRHS) [21,46]. Despite the somewhat contrasting taphonomic and diagenetic histories of *T. rex* MOR 1125 and *Edmontosaurus* bones from SRHS [46,51], both have been shown to retain endogenous cells, soft tissues [50,51], and collagen I protein [21,22,52]. More importantly, these specimens still share certain geochemical similarities indicative of diagenetic circumstances in common between these two sites that have evidently permitted long-term biomolecular preservation in fossil bones. We therefore now highlight these common themes in a preliminary attempt to constrain diagenetic pathways to molecular preservation.

Although our cumulative trace element data show that the coarse-grained nature of the entombing sand allowed MOR 1125 to experience overall greater chemical alteration than bones at SRHS [46], fossil specimens from both sites appear minimally altered at the elemental level. For example, they exhibit steeplydeclining REE profiles (Figure 4A, and Figures 1 and 2 of [46]), MREE concentrations which frequently drop below detection limit through the middle cortex (Data S1, and appendix A of [46]), and low ∑REE compared to other Cretaceous bones reported in the literature (see above). This similarity bears the obvious implication that minimal chemical alteration permits molecular preservation, as predicted by Trueman et al. [39]. Taphonomically, both assemblages were buried in lowland coastal settings, and the carcasses at each site underwent decay primarily in subaqueous environments ([49] and this study). These conditions could have permitted substantial trace element uptake, but instead it appears that burial soon after disarticulation followed by limited exposure to early-diagenetic pore fluids limited the magnitude of trace element uptake at each site. At SRHS, this was accomplished via rapid burial in a low-permeability mudstone which hindered pore fluid flow in partnership with partial encasement of select bones in early-diagenetic siderite concretions [46,50], and early cementation of the coarse sandstone at the MOR 1125 locality appears to have hampered diagenetic pore fluid flow in a similar fashion.

At the whole-bone level, specimens from both sites also exhibit significant enrichment in Sr compared to most other trace elements (as is common in bioapatitic fossils in general) [41,103], positive Y/Ho and (La/La*)_N_ anomalies (which are both common in fossil bones due to the slightly faster diffusivities of Y and LREE in bone) [101], high Sc enrichment (likely related to precipitation of secondary minerals under oxidizing conditions) [99,100], and slightly negative (Ce/Ce*)_N_ anomalies reflective of oxidizing diagenetic conditions (Table 1, and Table 1 of [46]). The last two of these findings agree with those of Wiemann et al. [109], who found that fossils from oxidizing depositional environments are more likely to yield endogenous soft tissues, despite the possibility of oxidative damage to biomolecules [35]. Indeed, in contrast to traditional views like those expressed by Eglinton and Logan [35], recent experiments by Boatman et al. [30] indicate that oxidizing conditions may actually promote biomolecular stabilization through free radical-induced inter- and intramolecular crosslinking.

Both MOR 1125 and bones from SRHS also exhibit clear signs of fractionation of REE during uptake from an early-diagenetic pore fluid, namely steeper concentration-depth profiles for LREE than HREE (Figure 4A, and Figure 2 of [46]) and shifting proportions of LREE and HREE by cortical depth (Figure 6B, and Figure 4 of [46]). Such fractionation patterns are commonly seen in fossil non-avian dinosaur bones possessing a thick cortex [40,43,65,116] due to a ‘filtering’ effect created by the thick rim of dense bone tissue, which can cause pore fluids to become relatively depleted in LREE by the time they reach the internal cortex [101]. Observation of such patterns in fossil bones is important, because they signify retention of early-diagenetic trace element signatures, which in turn demonstrates that late-diagenetic overprinting (which is nearly universal to some degree) [42,104] has neither completely obfuscated signatures imparted from the initial burial environment nor drastically altered the chemistry of the fossil specimen [36,48,80,117,118,119,120]. In other words, the marked fractionation trends seen in the femur of MOR 1125 and bones at SRHS demonstrate that the bulk of the REE inventory in each respective specimen derives from early-diagenetic uptake from a single pore fluid (rather than late-diagenetic overprinting), and that these bones avoided any significant interactions with other pore fluids during late diagenesis. This does not imply that the majority of the term of burial was ‘dry’, but rather that any exposure(s) to pore fluids after initial fossilization (such as phreatic groundwaters or recent vadose fluids) have not meaningfully influenced the chemistry of these fossils. Avoidance of protracted interactions with pore fluids through later phases of diagenesis thus appears to promote long-term biomolecular stability.

Interestingly, REE ternary diagrams, (Ce/Ce*)_N_ vs. (Pr/Pr*)_N_ plots, and anomaly profiles reveal considerable spatial heterogeneity in the degree of chemical alteration of fossil bones at both sites, especially within the internal cortex (Figure 5, Figure 8, Appendix A and Figures 3d, 6, and A.4 of [46]). Although it is not surprising that external cortices show more homogenous patterns of alteration (because they generally equilibrate with external pore fluids more quickly and to a greater degree) [83,101], recovery of endogenous soft tissues from such randomlyaltered bone tissue could be considered surprising. However, the magnitude of alteration is what likely matters more; as discussed above, the middle and internal cortices of each specimen exhibit lower trace element concentrations than the corresponding external cortex, signifying that even though patterns of alteration are comparatively more erratic internally, these internal regions still constitute the least-altered portion of dense bone tissue in each fossil. Therefore, based on these specimens, we suggest that unless signs of significant secondary diffusion from within the medullary cavity are encountered (e.g., Figure 11 of [120]), future paleomolecular sampling efforts should (where possible) target the middle and internal cortex of bones rather than the external cortex.

Finally, MOR 1125 and SRHS bones also each exhibit relatively high, flat profiles for Fe, Mn, and Ba (Figure 4C,D, and appendix A of [46]). We interpret these profiles to reflect homogenous distributions of minute, secondary mineral phases in the fossilized bone tissues (cf. [121]). Sustained peaks in any of these elements along a laser ablation transect could signify an unfavorable presence of more or larger crystals of secondary minerals (e.g., goethite, Mn oxides, and barite), which could be used alongside REE analyses as a potential screening tool to further direct paleomolecular sampling within the cortex of a fossil bone.

## 6. Conclusions

By considering the taphonomic and geochemical signatures of the femur of MOR 1125 and bones from SRHS in context with other recent actualistic and analytical studies cited in the Discussion above, we deduce the following about diagenetic pathways to molecular preservation:Our results strengthen the hypothesis that oxidizing depositional environments or, more specifically, oxidizing microenvironmental conditions during early diagenesis, can foster chemical reactions which stabilize bone cells, soft tissues, and their component biomolecules;In general, middle and internal cortical tissues usually experience less chemical alteration onaverage than the external cortex of any given fossil bone, implying that these interior regions should be the primary targets of future paleomolecular sampling;Diagenetic circumstances which restrict exposure to percolating pore fluids (i.e., burial in compact, fine-grained sediments, early/rapid cementation of sediments, encasement in early-diagenetic concretion) elevate molecular preservation potential by minimizing gross chemical alteration;Avoidance of protracted interactions with late-diagenetic pore fluids is almost certainly crucial to sustain long-term stability of biomolecules in a fossil bone.

These findings are intriguing, but it must be remembered that our conclusions remain based on data from only two Cretaceous localities in the same geologic formation, physiographic region, and current climatic regime (the Hell Creek Formation in the U.S. Western Interior). To fully elucidate molecular preservation mechanisms and the taphonomic circumstances which facilitate them, traditional taphonomic and trace element analyses must be conducted alongside paleomolecular assays (e.g., immunoassays or proteomics) at additional localities of varied ages which yield bones preserved under widelydiverse paleoenvironmental and diagenetic contexts.

## Figures and Tables

**Figure 1 biology-10-01193-f001:**
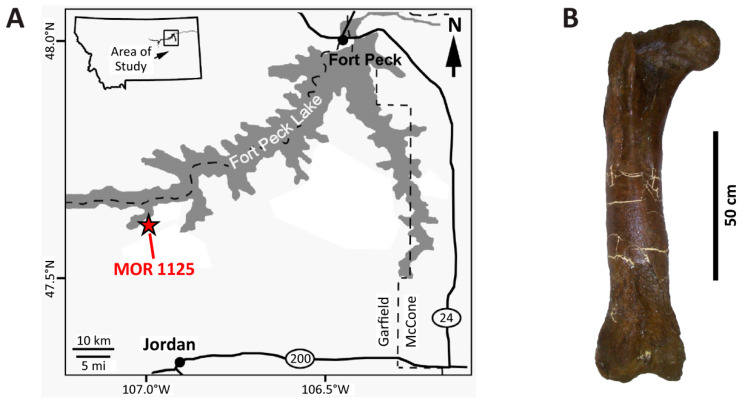
(**A**) Map showing the location of the MOR 1125 quarry in Garfield County, Montana. (**B**) Right femur of MOR 1125 examined in this study. Map modified from [54] and bone photograph modified from [55], each under CC BY-4.0 licenses.

**Figure 2 biology-10-01193-f002:**
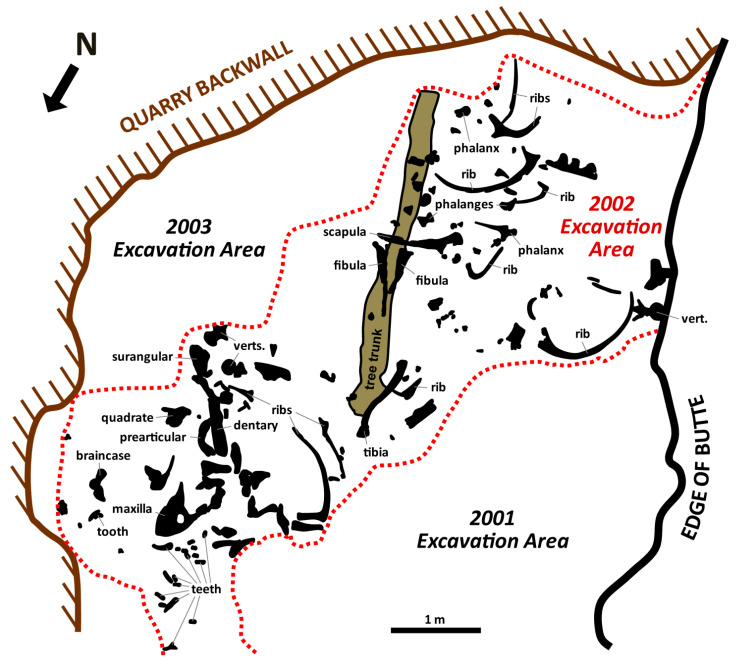
Quarry map from the 2002 field season at the MOR 1125 quarry. Only bones discovered during that field season are shown. Select identifiable bones are identified as labeled. Although the large tree trunk was found above all the bones in the central region of the quarry, it is shown behind them to allow the bones to be better seen. Scale bar as indicated.

**Figure 3 biology-10-01193-f003:**
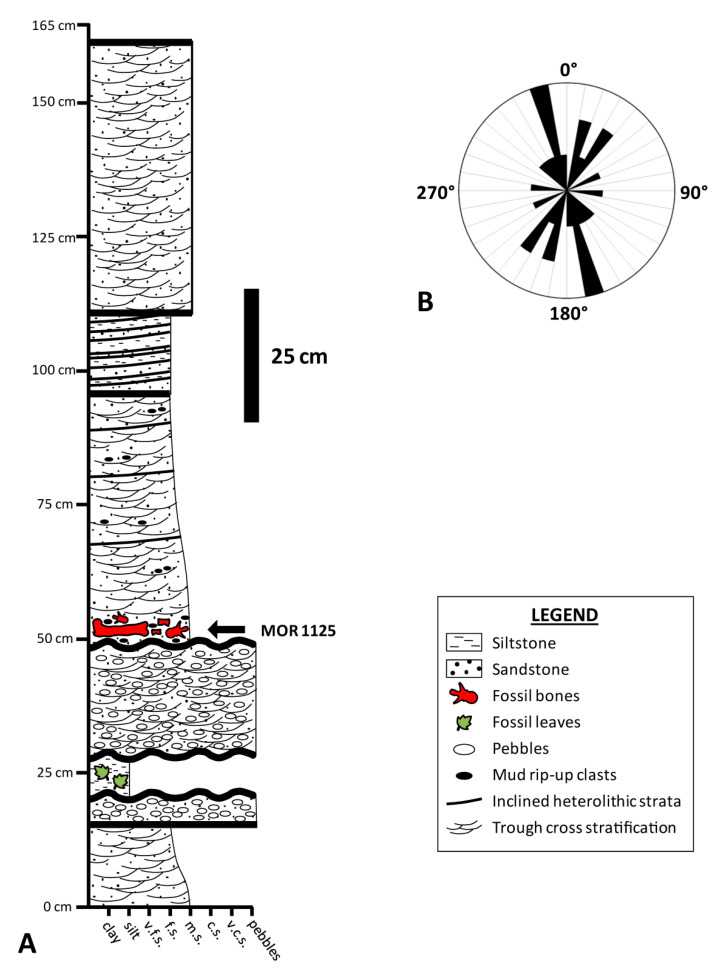
(**A**) Stratigraphic section taken within the MOR 1125 quarry. (**B**) Rose diagram of MOR 1125 bone orientations based on data from 14 long bones collected during the 2002 field season. Presented as an arithmetic plot with 10° bins. Abbreviations: c.s.—coarse sand; f.s.—fine sand; m.s.—medium sand; v.c.s.—very coarse sand; v.f.s.—very fine sand. Scale bar for (**A**) as indicated.

**Figure 4 biology-10-01193-f004:**
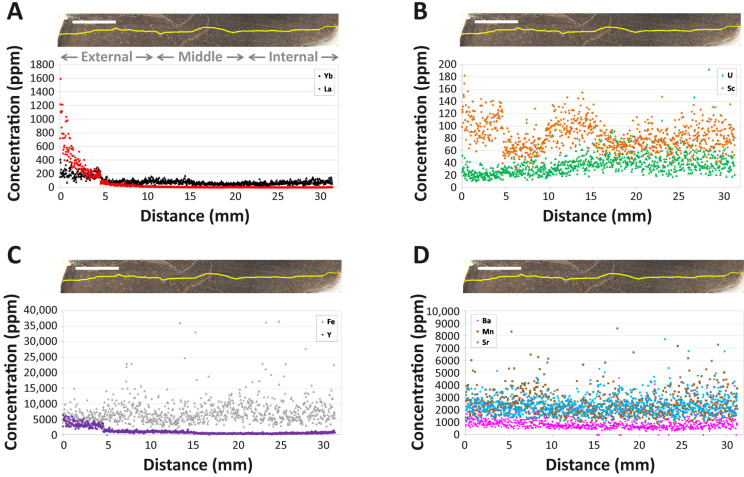
Intra-bone REE concentration gradients of various elements in the femur of MOR 1125. (**A**) Lanthanum (La) and ytterbium (Yb). (**B**) Scandium (Sc) and uranium (U). (**C**) Iron (Fe) and yttrium (Y). (**D**) Barium (Ba), manganese (Mn), and strontium (Sr). Note the different concentration scales for each panel. The laser track is denoted by the yellow line in each bone cross section. Gray text labels in (**A**) span the approximate regions considered as the ‘external’, ‘middle’, and ‘internal’ cortices. Scale bars, in white over bone images, each equal 1 mm.

**Figure 5 biology-10-01193-f005:**
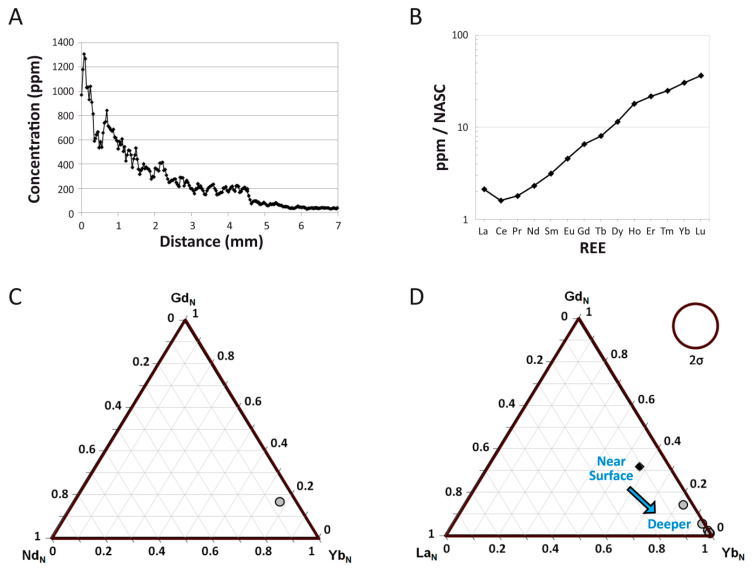
REE composition of the femur of MOR 1125. (**A**) Three-point moving average profile of La concentrations in the outermost 7 mm of the bone. (**B**) Average NASC-normalized REE composition of the fossil specimen as a whole. (**C**,**D**) Ternary diagrams of NASC-normalized REE. (**C**) Average composition of the bone. (**D**) REE compositions divided into data from each individual laser transect (~5 mm of data each). Compositional data from the transect that included the outer bone edge is denoted by a dark diamond; all other internal transect data are indicated by gray circles. The 2σ circle represents two standard deviations based on ±5% relative standard deviation.

**Figure 6 biology-10-01193-f006:**
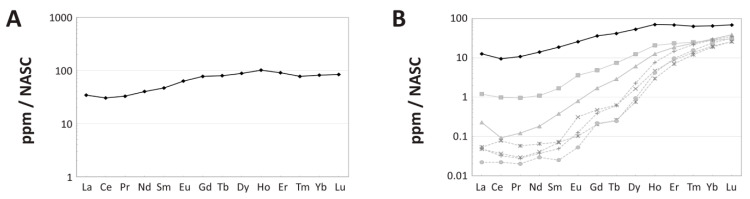
Spider diagrams of intra-bone NASC-normalized REE distribution patterns within the femur of MOR 1125. (**A**) Average composition of the outermost 250 µm of the cortex, demonstrating a similar magnitude of relative LREE depletion in the outermost cortex as seen in the bone as a whole (Figure 5B). (**B**) Variation in compositional patterns by laser transects. The pattern which includes the external margin of the bone is shown in black, those from deepest within the bone by dotted, lightgray lines, and all other analyses in between by solid, darkgray lines.

**Figure 7 biology-10-01193-f007:**
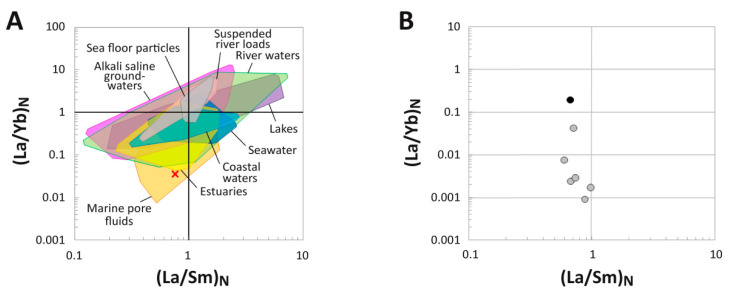
(La/Yb)_N_ and (La/Sm)_N_ ratios of the femur of MOR 1125. (**A**) Comparison of the whole-bone average (La/Yb)_N_ and (La/Sm)_N_ ratios of the fossil to ratios from various environmental waters and sedimentary particulates. Literature sources for environmental samples are provided in the Appendix A. (**B**) REE compositions of individual laser transects expressed as NASC-normalized (La/Yb)_N_ and (La/Sm)_N_ ratios. The transect including the external bone margin is denoted by the black symbol, whereas all other (internal) transects are represented by gray symbols.

**Figure 8 biology-10-01193-f008:**
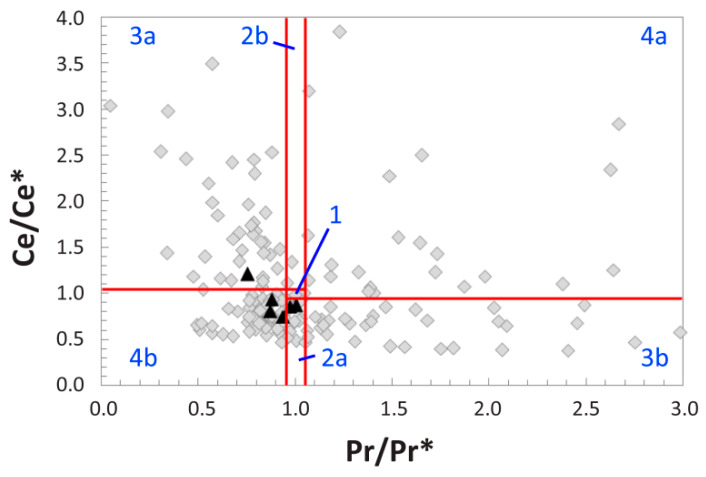
(Ce/Ce*)_N_ vs. (Pr/Pr*)_N_ plot (after [66]) of five-point averages along the transect across the cortex of MOR 1125 recorded by LA-ICPMS. Separate fields (labeled by blue text) are as follows: 1, neither Ce nor La anomaly; 2a, no Ce and positive La anomaly; 2b, no Ce and negative La anomaly; 3a, positive Ce and negative La anomaly; 3b, negative Ce and positive La anomaly; 4a, negative Ce and negative La anomaly; 4b, positive Ce and positive La anomaly. Measurements from the outer 1 mm of the external cortex are plotted as black triangles, and all measurements from deeper within the bone are plotted as grey diamonds. (Ce/Ce*)_N_ and (Pr/Pr*)_N_ anomalies, comparing observed (Ce, Pr) versus expected (Ce*, Pr*) concentrations of each element, are calculated as in the Materials and Methods section of the text.

**Figure 9 biology-10-01193-f009:**
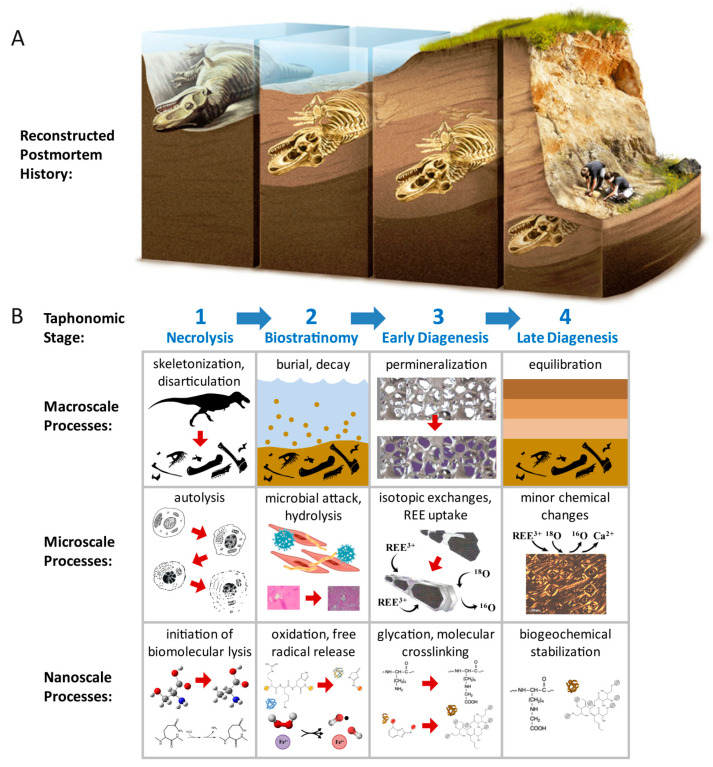
Reconstruction of the taphonomic and diagenetic history of *Tyrannosaurus rex* MOR 1125. (**A**) Generalized postmortem history of the carcass, *sans* disarticulation. Reprinted in modified form with permission from ref. [108]. Copyright 2016 Porto Editora. (**B**) Synopsis of macroscale, microscale, and nanoscale (molecular level) processes inferred to have taken place during each taphonomic stage in the decay of this specimen, as portrayed in (**A**). Nanoscale processes of biomolecular decay and stabilization based on propositions by [29,30,109,110]. Black *Tyrannosaurus* silhouette by Scott Hartman, www.phylopic.org (accessed on 8 August 2021), CC BY-NC-SA-3.0. Autolysis, REE uptake, biomolecular lysis, and glycation images each respectively modified, with permission, from [38,111,112,113,114]. Hydrolysis image modified from [115], and oxidation and crosslinking images modified from [109], each under CC BY-4.0 licenses.

**Table 1 biology-10-01193-t001:** Average whole-bone trace element composition of the right femur of *Tyrannosaurus rex* MOR 1125. Numbers presented are averages of all transect data acquired across the cortex. Iron (Fe) is presented in weight percent (wt. %); all other elements are in parts per million (ppm). Absence of (Ce/Ce*)_N_, (Pr/Pr*)_N_, (Ce/Ce**)_N_, and (La/La*)_N_ anomalies occurs at 1.0. (Ce/Ce*)_N_, (Pr/Pr*)_N_, (Ce/Ce**)_N_, and (La/La*)_N_ anomalies, comparing observed (Ce, Pr, La) versus expected (Ce*, Pr*, Ce**, La*) concentrations of each element, are calculated as in the Materials and Methods section of the text. The Y/Ho value reflects this mass ratio.

Element	Concentration
Sc	83.43
Mn	2439
Fe	0.73
Sr	2386
Y	1102
Ba	888
La	66.48
Ce	107.47
Pr	14.24
Nd	63.72
Sm	17.65
Eu	5.45
Gd	34.38
Tb	6.88
Dy	63.45
Ho	18.82
Er	74.13
Tm	12.51
Yb	93.69
Lu	16.68
Th	0.13
U	37.77
∑REE	596
(Ce/Ce*)_N_	0.82
(Pr/Pr*)_N_	0.92
(Ce/Ce**)_N_	1.26
(La/La*)_N_	2.83
Y/Ho	58.55

**Table 2 biology-10-01193-t002:** Summary of the REE composition of the right femur of *Tyrannosaurs rex* MOR 1125. Qualitative ∑REE content is based on the value shown in Table 1 (596 ppm) in comparison to values from other Mesozoic bones (as listed in the main text). Abbreviations: DMD—double medium diffusion *sensu* [44]; LREE—light rare earth elements.

Clear DMD Kink for LREE?	Relative Noise in Outer Cortex for La	REE Suggest Flow in Marrow Cavity?	Relative ∑REE Content (Whole Bone)	Relative U Content (Whole Bone)	Relative Porosity of the Cortex
No	Low	No	Low	Low	Low

## Data Availability

All data generated by this study are available in this manuscript and the accompanying Appendix A.

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
