# Peer review of "Taphonomic and Diagenetic Pathways to Protein Preservation, Part I: The Case of Tyrannosaurus rex Specimen MOR 1125"

_biology, 2021, doi:10.3390/biology10111193_

Round 1

Reviewer 1 Report

This work focuses on the study of diagenetic processes that may allow the preservation of molecular content in fossil remains. For this purpose, the results of the taphonomic study of the remains of a Tyrannosaurus rex individual, specifically specimen MOR 1125, are presented. The analysis focuses on biostratinomic aspects and, mainly, on fossildiagenetic aspects. Specifically, they perform an analysis of the REE present in the femur. This skeletal element has been the subject of a previous study showing that it preserved evidence of soft tissues and endogenous proteins.

In my opinion this work is robust and detailed, both in data analysis and interpretation. The structure and figures of the results are too similar to those already published by the authors in other works (see Ullmann et al., 2020 in Geochim. Cosmochim. Ac.), although it does not detract from the information provided.

In this sense, the graphic information provided is adequate, although a location map of the Hell Creek Formation and perhaps some photographs of the specimen analyzed would have been interesting.

The reconstruction of the taphonomic history (Figure 8A), although it is a reproduction and responds to a general scheme, should include the skeletal disarticulation described by the authors, since such disarticulation occurs prior to burial and could lead to confusion.

I must emphasize that the summary of data included in Figure 8B is really explanatory, including the different stages of diagenesis, the levels of analysis (macro, micro and nanoscale) and the processes involved.

The discussion integrates well all the data provided and some really interesting conclusions are reached that can be extrapolated to other fossil assemblages.

The bibliography is adequate and incorporates the latest research in the field of study. The information included in the supplementary material is really interesting.

The title includes "part I", which suggests that there is a second work (part II), linked to the proposals described in the last paragraph of the manuscript?

In my opinion this work can be published in its present form. I believe that it provides methodological solutions, using traces element analysis, that can be applied in future work on molecular conservation. In addition, I believe that it can contribute substantially to the understanding of taphonomic processes, both in early and late diagenesis, that are involved in molecular conservation mechanisms.

Reviewer 2 Report

The manuscript by Ullmann et alii represents an exquisitely crafted example of cutting-edge taphonomic research. Although my experience with paleoproteomics is very limited, and I am not a geochemist, the submitted paper is excellent as far as I can judge.

Some suggestions for minor revisions are as follows:

- Introduction: for the general readership, you may clarify that protein preservation is something different from the preservation of soft proteinaceous structures like horn (e.g. Armitage et al., 2013), baleen and hair (e.g. Ji et al., 2002) via phosphatization or carbonification processes.

- Fig. 1, and caption: using the term ‘log’ here (and several times below) may be ambiguous – someone could understand that you are referring to a measured stratigraphic log… What about ‘tree trunk’?

- l. 173-183: based on the main text alone, it is not clear very what the ‘butte’ refers to, and what kind of the stratigraphic relationship occur between the Hell Creek Fm and the other formations that are referred to herein. I suggest some rephrasing for the sake of maximum clarity.

- l. 205: change ‘taphonomically’ to ‘macroscopically’.

- Fig. 5, caption: I see more than one dotted, light gray line, maybe the figure caption should be revised?

- l. 435: please use km instead of miles.

- l. 436-437: I am not sure to understand why the presence of wood and leave remains are especially indicative of a burial environment near the coast…

- l. 478: change ‘decomposition’ to ‘skeletonization’ or ‘defleshing’

- l. 637: I guess that could be the third such study – for example, Bosio et al. (2021) provided a thorough geochemical characterization of marine mammal bones from the Miocene Pisco Formation, where protein preservation is also demonstrated (Boskovic et al., 2021).

- l. 690: I think that the recent paper by Bosio et al. (2021) should be profitably cited here. (By the way, it may be mentioned in the Introduction, l. 110-115, as well.)

- l. 727-729: perhaps this conclusion is a bit risky! I would personally be content with stating that your new results strengthen the notion that the preservation of proteines is possible in oxidizing conditions (or something to the same effect).

Looking forward for reading this nice piece of research published on Biology very soon!

Bests,

the reviewer

Armitage, M. H., & Anderson, K. L. (2013). Soft sheets of fibrillar bone from a fossil of the supraorbital horn of the dinosaur Triceratops horridus. Acta histochemica, 115(6), 603-608.

Bosio, G., Gioncada, A., Gariboldi, K., Bonaccorsi, E., Collareta, A., Pasero, M., ... & Bianucci, G. (2021). Mineralogical and geochemical characterization of fossil bones from a Miocene marine Konservat-Lagerstätte. Journal of South American Earth Sciences, 105, 102924.

Boskovic, D. S., Vidal, U. L., Nick, K. E., Esperante, R., Brand, L. R., Wright, K. R., ... & Siviero, B. C. (2021). STRUCTURAL AND PROTEIN PRESERVATION IN FOSSIL WHALE BONES FROM THE PISCO FORMATION (MIDDLE-UPPER MIOCENE), PERU. Palaios, 36(4), 155-164.

Ji, Q., Luo, Z. X., Yuan, C. X., Wible, J. R., Zhang, J. P., & Georgi, J. A. (2002). The earliest known eutherian mammal. Nature, 416(6883), 816-822.

Reviewer 3 Report

This manuscript is so well done in its careful scholarship, clear writing and thorough documentation of the taphonomical, geological and biomolecular preservation context, that I have very little to add or critique.  The authors made a solid case in providing clues regarding the preservation mechanisms of biomolecules and the taphonomic circumstances that facilitate them at least for some depositional environments. Therefore, I recommend the publication of the manuscript in its present form.

Author Response

We sincerely thank Reviewer 3 for their kind compliments of our manuscript and for taking the time to review it. Since Reviewer 3 requested no changes, we have made no changes to the manuscript in response to this reviewer's comments.